# Dealing with Noisy Data in Federated Learning: An Incentive Mechanism with Flexible Pricing

## ABSTRACT

Federated Learning (FL) has emerged as a promising training framework that enables a server to effectively train a global model by coordinating multiple devices, i.e., clients, without sharing their raw data. Keeping data locally can ensure data privacy, but also makes the server difficult to assess data quality, leading to the noisy data issue. Specifically, for any given taring task, only a portion of each client's data is relevant and beneficial, while the rest may be redundant or noisy. Training with excessive noisy data can degrade performance. Motivated by this, we investigate the limitations of existing studies and develop an incentive mechanism with flexible pricing tailored for noisy data settings. The insight lies in mitigating the impact of noisy data by selecting appropriate clients and incentivizing them to clean their data spontaneously. Further, both rigorous theoretical analysis and extensive simulations compared with state-of-the-art methods have been well-conducted to validate the effectiveness of the proposed mechanism.

## CCS CONCEPTS

• **Computing methodologies** → *Distributed computing methodologies.*

## KEYWORDS

Federated learning, noisy data, incentive mechanism

### ACM Reference Format:

. 2018. Dealing with Noisy Data in Federated Learning: An Incentive Mechanism with Flexible Pricing. In *Proceedings of Make sure to enter the correct conference title from your rights confirmation emai (Conference acronym 'XX).* ACM, New York, NY, USA, 10 pages. https://doi.org/XXXXXXX.XXXXXXX

## 1 INTRODUCTION

In recent years, Federated Learning (FL) has emerged as a promising decentralized training framework. It leverages the power of multiple devices, referred to as clients, to collectively train a global model without sharing clients' local data, thereby ensuring both efficiency and privacy [16]. Consequently, FL has been extensively studied and applied in various fields [3, 4, 29].

Typically, a FL system incorporates two main components: a *server* and multiple *clients*. The server trains a global model by iteratively coordinating clients over finite rounds. At each round, clients perform local training, produce local models, and communicates them back to the server for aggregation. Throughout this

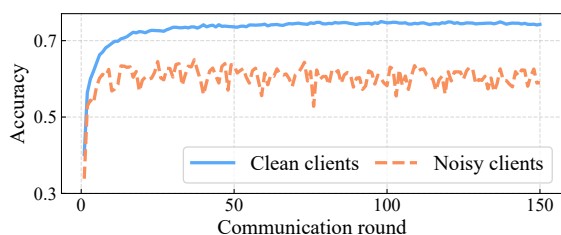

**Figure 1: Components of defined noise score.**

process, raw data remains stored locally on the clients, thereby preserving data privacy. This, however, also prevents the server from assessing the data quality, giving rise to the *noisy data issue.*

More precisely, given a specific task like handwritten digit recognition [8], the printed digits included in each client's local dataset can be regarded as the redundant and noisy data. Evidently, including such noisy data, especially when it is associated with the same labels as relevant data, will mislead and manipulate the training process unpredictably, significantly degrading the FL performance as depicted in Fig. 1.

This noisy data issue was first proposed and studied by Tuor *et al.* [26]. They introduced a centralized benchmark model trained on a small, task-specific dataset to select relevant data for each client during local training. In contrast, Nagalapatti *et al.* [20] developed FLRD, which allows clients to train their own relevant data selection models, further facilitating local training. Different from these methods [20, 26] that focus on the local training phase, Li *et al.* [10] considered the aggregation phase and proposed a learning-based reweighting approach that adjusts the weight for each training sample. However, we argue that existing studies still have notable limitations in practical scenarios, say, they all relied on auxiliary, task-specific datasets to tackle the issue, incurring additional training costs and reduced generality.

Motivated by this, we explore a novel perspective to address the noisy data issue: an incentive mechanism approach. An incentive mechanism typically comprises two phases: the client selection phase that picks clients for local training and the pricing phase that determines payments to compensate clients' expense [21]. Its feasibility and effectiveness lie in alleviating the negative impact of noisy data by selecting clients with low noise and high complementarity during the selection phase, and by incentivizing clients to clean their noisy data spontaneously by paying suitably in the pricing phase.

Designing such a mechanism tailored for noisy data issues presents several specific challenges. One essential step before client selection is to detect and measure the noise level of each client for various training tasks. To enhance generality, we avoid relying on prior knowledge, such as the auxiliary datasets used in [20, 26]. This makes accurately detecting noisy data becomes even more difficult. Thus, the first challenge emerges as *designing a noise detection*

*policy that effectively balances the trade-off between accuracy and uncertainty.*

Merely detecting the noise level is insufficient, especifically when detection accuracy cannot be ensured without prior knowledge. To this end, we further tackle the noisy data issue in the client selection phase. In addition to considering the noisy levels of clients, we also focus on leveraging the complementarity between clients to counteract the negative impact of noisy data on local training. Unfortunately, clients' complementarity is unknown in advance, making it challenging to choose appropriate client sets. One possible solution is to iteratively explore and try different client sets over rounds and observe their actual training utility. However, one cannot consistently explore new client sets; it is also important to exploit and choose those client sets that have performed well previously, to facilitate the final training performance. Therefore, *the second challenge is to select low-noise clients while balancing the trade-off between exploration and exploitation.*

After selection, we tend to pay these selected clients for two goals. One is to cover their expense to motivate them join FL, i.e., individual rationality. Another one is further incentivizing clients to clean their noisy data spontaneously. Existing incentive mechanisms can successfully achieve the first goal by paying clients based on their costs, which are submitted by themselves via the reverse-auction framework [14, 18, 27]. They also ensure clients to submit their true costs rather than lies, i.e., truthfulness. However, these existing mechanisms fail to further achieve the second goal as they cannot accurately control the produced payments. Hence, *the third challenge is to price accurately and flexibly, thereby incentivizing clients to clean their noisy data while enforcing truthfulness.*

To overcome these challenges, we develop a novel Flexible Pricing based Incentive mechanism tailored for Noisy FL settings (FPIN). For the first challenge, FPIN presents a noise detection policy, which allows the server to measure and quantify the noise level of a client based on the discrepancy between its submitted local model and the aggregated global model. This policy merely relies on each client's local model rather than an auxiliary dataset, enhancing generality of FPIN. For the second challenge, FPIN includes a selection policy, which utilizes a combinatorial bandit based online learning method to effectively balances exploring unknown and potential client sets with exploiting low-noise and high-complementarity client sets. Actually, this policy iteratively gains useful knowledge from making mistakes over rounds, ultimately facilitating the training performance. For the third challenge, FPIN contains a pricing policy, which can flexibly and dynamically control the payment produced for selected clients based on their noise levels. This thereby encourages clients to clean their own data while achieving both truthfulness and individual rationality. Finally, FPIN's effectiveness is validated through both theoretical analysis and experimental simulations. Our contributions are summarized as follows:

- **Problem.** As far as we know, we are the first to address the noisy data issue from an incentive prospective. The insight lies in selecting low-noise clients, utilizing complementarity between them, and encouraging them by suitable payments.
- **Method.** We carefully study the impact posed by noise data and develop a Flexible Pricing based Incentive mechanism

tailored for Noisy FL settings (FPIN), including a noise detection policy, a client selection policy, and a pricing policy.
- **Analysis.** We provide crucial guarantees of FPIN through theoretical analysis. It includes selection regret bound, truthfulness, individual rationality, noise robustness, and convergence rate of the whole training process.
- **Simulation.** We conduct extensive experimental simulations based on real-world datasets compared with well-known benchmarks. The results align with our theoretical findings and illustrate the effectiveness of FPIN.

The rest of the paper is organized as follows. We review related works in Section 2, introduce the system model, and formulate the analyzed problems in Section 3.We design our framework in Section 4. The effectiveness of the framework is evaluated in Section 5 theoretically and in Section 6 numerically. The paper is concluded in Section 7. Appendices are shown in Section 8.

## 2 RELATED WORK

### 2.1 Noise in FL

The noisy label problem has been widely analyzed in federated learning to effectively improve the robustness of the system. In the beginning, Tuor et al.[26] proposed a distributed method that uses a small benchmark model to evaluate the relevance of data samples at each client, and then only selects the relevant data to participate in the federated learning process. Unlike previous approaches, FLRD introduced by Nagalapatti et al. [20] allows clients to build their own models for relevant data selection, leading to more efficient local training. Li et al.[9] introduced FedDiv, which extracts knowledge from all clients to facilitate federated noise filtering. Previous research has mainly focused on the local training phase, emphasizing client-side model optimization while often overlooking the challenges of global model aggregation and the effects of noisy clients on overall performance. Focusing on the aggregation phase, Li et al. [10] proposed a learning-based reweighting approach that modifies the weight assigned to each training sample. Fang et al.[2] proposed RHFL that addresses label noise by aligning heterogeneous model feedback using public data, applying a noise-tolerant loss function, and implementing a client confidence reweighting scheme for adaptive collaboration. Nonetheless, we argue that existing studies have considerable limitations in practical contexts, primarily due to its reliance on auxiliary, task-specific datasets to tackle the challenges, leading to higher training costs and diminished applicability.

### 2.2 Incentive Machanism in FL

Incentive mechanisms based on game theory, auction theory, and others have been extensively studied in federated learning. Pan et al.[21] proposes a new incentive mechanism for graph federated learning, addressing hclientful and delayed agent contributions by introducing an agent valuation function based on gradient alignment and graph diversity. Murhekar et al.[17] models a collaborative FL framework, introducing a budget-balanced mechanism to maximize agents' welfare, along with a protocol FedBR-BG utilizing best response dynamics. Wu et al.[27] presents an incentive-aware algorithm that offers differentiated training-time model rewards

to clients in federated learning, addressing challenges with post-training incentives and ensuring optimal model recovery. Zhang et al.[30] proposed RRAFL, a federated learning incentive mechanism based on reputation and reverse auction, which selects participants through a reputation assessment that indirectly reflects their data quality and reliability. Lu et al.[14] presents MAGFL, a Multi-attribute Auction-based Grouped Federated Learning scheme that clusters clients, evaluates group quality, distributes economic rewards, and incorporates Adam operations to accelerate convergence. However, these existing mechanisms cannot further incentivize clients to voluntarily clean their noisy data because they cannot accurately control the payments generated.

## 3 PRELIMINARIES

### 3.1 Reverse-auction based FL System

We consider a reverse-auction based FL system, including a cloud server that acts as a buyer, denoted by $\mathcal{S}$, and $N$ distributed clients as the sellers, denoted by $[N]$. Products are training services that clients can provide for server $\mathcal{S}$. Each client $i \in [N]$ maintains a local model denoted by vector $w_{i,t}$, where $t \in [T]$ represents the $t$-th round given total $T$ discrete communication rounds of this system. Also, each client is associated with a local dataset $\mathcal{D}_i = \{(x_j, y_j) : j \in [M_i]\}$, where $y_j$ is the ground-truth label with respect to $x_j$ and $M_i = |\mathcal{D}_i|$. In practical scenarios, there is several noise contained in $\mathcal{D}_i$, i.e., the data irrelevant with the given training task, which is represented by $\mathcal{D}_i' \subseteq \mathcal{D}_i, \forall i \in [N]$. We then provide the specific workflow of the reverse-auction based FL system.

**Cost submission.** At the beginning of each round $t$, server $\mathcal{S}$ solicit costs of all clients for subsequent pricing. Clients then upload their costs, denoted by $c_{i,t}$, to represent their expense like computational consumption and communication overhead [13]. **Client selection.** Receiving costs of clients, server $\mathcal{S}$ chooses a client set $I_t$ out of total $N$ clients according to both their costs and previous feedback on training contributions. **Local training.** These selected clients $I_t$ then start local training. The loss of client $i$ on a specific labeled example $d = (x, y)$ is denoted as $f_i(w_{i,t}, d)$. The average loss over local dataset $\mathcal{D}_i$ is denoted as

$$F_i(w_{i,t}, \mathcal{D}_i) = (1/|\mathcal{D}_i|) \sum_{d \in \mathcal{D}_i} f_i(w_{i,t}, d) \quad (1)$$

and the local training goal of client $i$ is to find a model $w_{i,t}$ that yields an acceptably small average loss,

$$w_{i,t} = \arg\min_w F_i(w, \mathcal{D}_i). \quad (2)$$

**Global aggregation.** Local models $w_{i,t}$ derived in Eq. 2 are then uploaded by clients in $I_t$ to server $\mathcal{S}$ and are aggregated to a global model as $w_t = \sum_{i \in I_t} p_i w_{i,t}$, where the weight $p_i = |\mathcal{D}_i| / \sum_{k \in I_t} |\mathcal{D}_k|$. This aggregated model $w_t$ is subsequently downloaded to each client. **Payment determination.** Afterward, sever $\mathcal{S}$ determines payments for selected clients based on their costs and contributions, denoted by $p_{i,t}, \forall i \in I_t$. After the five phases mentioned above, round $t$ terminates and the next round $t+1$ starts. All FL rounds ultimately end as the global model $w_T$ convergences at round $T$. Therefore, the final training goal of FL is to find a global model $w_T$ such that

$$w_T = \arg\min_w \sum_{i \in I_t} p_i F_i(w, \mathcal{D}_i). \quad (3)$$

We primarily focus on client selection and payment determination phases of the FL system in this paper, which are modeled as follows.

### 3.2 Selection Model

The insight of the selection phase is to sample clients with both low noise and high contribution to training. We define the noise score $l_i = l(\mathcal{D}_i)$ to represent noise levels of client $i \in [N]$ and which will be precisely quantified in subsequent sections. A higher value of $l_i$ indicates a greater noise level. Afterward, to measure clients' contributions to training, the optimal approach is to use their importance $|\mathcal{D}_i| \sqrt{\frac{1}{|\mathcal{D}_i|} \sum_{d \in \mathcal{D}_i} \|\nabla f_i(w_{i,t}, d)\|^2}$, where $\nabla f_i(w_{i,t}, d)$ is the L2-norm of the gradient of a given sample $d \in \mathcal{D}_i$ [6]. However, this approach is impractical as calculating this importance introduces too much extra computational time. Instead, we utilize a pragmatic variant of this importance to represent the statistical utility inspired by [5, 7]. We further consider the noise level and formally define the statistical utility in Definition 1. The insight is a larger gradient norm intuitively yields a higher loss. Also, the selection policy is given in Definition 2.

**DEFINITION 1 (STATISTICAL UTILITY).** *Reflecting both the importance and noise level, the statistical utility of a client $i \in [N]$ at round $t \in [T]$ is formally represented by*

$$u_{i,t} = \frac{|\mathcal{D}_i|}{l(\mathcal{D}_i)} \sqrt{\frac{1}{|\mathcal{D}_i|} \sum_{d \in \mathcal{D}_i} f_i(w_{i,t}, d)^2}. \quad (4)$$

*All statistical utilities of client $i$ up to round $t$ can be denoted by a sequence $U_{i,t} = \{u_{i,\tau} : i \in I_\tau, \tau \in [1:t]\}$, where $I_\tau$ represents the client set selected at communication round $\tau$.*

**DEFINITION 2 (SELECTION POLICY).** *Given the cost set $C_t = \{c_{i,t} : i \in [N]\}$, the utility set $\mathcal{U}_t = \{u_{i,t} : i \in [N]\}$, and the cardinality constraint $K$, a selection policy $\pi_s$ assists server $\mathcal{S}$ in sampling a client set $I_t$ to optimize the global model, i.e., $\pi_s(C_t, \mathcal{U}_t, K) = I_t$.*

### 3.3 Pricing Model

In the system, we assume that all clients are rational and selfish [27], indicating that each client $i \in [N]$ may declare a false cost $c_{i,t}' \neq c_{i,t}$ to get more payments. This results in unfair competition among clients, thereby degrading the training performance. To prevent such strategic behaviors, cover clients' expense, and incentivize clients to clean noise, the pricing policy $\pi_p$ should be designed to achieve truthfulness, individual rationality, and noise robustness. Formally, we define the pricing policy in Definition 3 and these properties in Definitions 4-6.

**DEFINITION 3 (PRICING POLICY).** *The pricing policy $\pi_p$ is utilized by server $\mathcal{S}$ to determine the payment for each client in $I_t$, i.e., $\pi_p(c_{i,t}, C_{-i,t}, \mathcal{U}_t, \kappa) = p_{i,t}, \forall i \in I_t$, where $C_{-i,t} = C_t \setminus \{c_{i,t}\}$.*

**DEFINITION 4 (TRUTHFULNESS).** *The pricing policy $\pi_p$ achieves truthfulness if for any fake cost $c_{i,t}' \in \mathbb{R}$ and $c_{i,t}' \neq c_{i,t}$, it holds that*

$$\pi_p(c_{i,t}, C_{-i,t}, \mathcal{U}_t, K) \geq \pi_p(c_{i,t}', C_{-i,t}, \mathcal{U}_t, K), \forall t \in [T]. \quad (5)$$

*This implies that being truthful is the dominant strategy for clients.*

**DEFINITION 5 (INDIVIDUAL RATIONALITY).** *The pricing policy $\pi_p$ is individually rational if for any client $i \in [N]$, it holds that*

$$\pi_p(c_{i,t}, C_{-i,t}, \mathcal{U}_t, K) \geq c_{i,t}, \forall t \in [T]. \quad (6)$$

*This ensures that the payment is sufficient to cover clients' expense.*

**DEFINITION** 6 (NOISE ROBUSTNESS). *The pricing policy $\pi_p$ achieves noise robustness in noisy FL if for any client $i \in [N]$, it holds that*

$$\pi_p(c_{i,t}, C_{-i,t}, \mathcal{U}_t, K) - c_{i,t} \leq l(\mathcal{D}_i), \forall t \in [T]. \tag{7}$$

*This indicates that the client with low noise levels can obtain a better award in addition to the part covering the cost.*

## 3.4 Problem Formulation

The key problem involved in selection and pricing phases is to develop policy $\pi_s$ and policy $\pi_p$. For $\pi_s$, it aims to select clients with high statistical utilities at each round iteratively, further maximizingz the expected cumulative utility $\mathbb{E}[U_{\pi_s}(T)]$ over total $T$ rounds. This problem is referred to as the *noisy client selection problem*, i.e.,

$$\text{Maximize} : \mathbb{E}[U_{\pi_s}(T)] = \mathbb{E}[\sum_{t \in [T]} \sum_{i \in [N]} x_{i,t} u_{i,t}], \tag{8}$$

$$\text{Subject to} : \quad x_{i,t} \in \{0, 1\}, \forall i \in [N], t \in [T], \tag{9}$$

$$|I_t| = K, I_t \subseteq [N]. \tag{10}$$

In Eqs. 8 and 9, $x_{i,t}$ is an binary indicator denoting whether a client $i$ is selected at round $t$, where 1 for selected and 0 for not selected. $I_t \subseteq [N]$ is the client set selected at each round $t$, i.e., $x_{i,t} = 1, \forall i \in I_t$. Eq. 10 indicates the cardinality constraint. It can be observed that maximizing the cumulative utility over $T$ rounds is substantially equivalent to minimizing its regret $\mathcal{R}_{\pi_s}(T)$, which is defined as the utility difference between policy $\pi_s$ and the optimal policy $\pi_s^*$,

$$\mathcal{R}_{\pi_s}(T) = w_{\pi_s^*}(T) - \mathbb{E}[w_{\pi_s}(T)], \tag{11}$$

where $w_{\pi_s^*}(T) = \max_{I \subseteq [N]:|I| = K} \sum_{t \in [T]} \sum_{i \in I} u_{i,t}$ is the cumulative utility of consistently selecting the best $K$-size client set. For policy $\pi_p$, it aims to pay clients flexibly and accurately in order to achieve truthfulness, individual rationality, and noise robustness. This is referred to as the *flexible pricing problem*.

## 4 MECHANISM DESIGN OF FPIN

We describe here the details of Flexible Pricing based Incentive mechanism tailored for Noisy FL settings (FPIN).

## 4.1 Noise Level Detection

Accurately detecting the noise levels $l(\mathcal{D}_i)$ of each client is crucial for the following selection and pricing phases. However, as we mentioned above, previous studies either rely on auxiliary datasets [20, 26] or require all clients to join a pre-training process for noise detection [28], leading to impracticality for FL applications.

We aim to explore a practical approach for identifying clients' noise levels. Pre-simulations revel that, during the training process, clients with high noise levels consistently exhibit a local model that diverges more significantly from the global model compared to low-noise and clean clients. Based on these findings, we let the noise level $l(\mathcal{D}_i)$ for client $i$ be proportional to the discrepancy of the aggregated global model and the local model. Specifically, $l(\mathcal{D}_i) \propto \|w_t - w_{i,t}\|^2$, where $\|w_t - w_{i,t}\|^2$ represents the Euclidean distance between two model parameters $w_t$ and $w_{i,t}$.

Yet, only the model discrepancy cannot describe the noise level sufficiently. Studies on deep learning have revealed two phases of the model evolution: The former is dimensionality compression that captures underlying data distribution, while the latter is dimensionality expansion that enables the model to fit clean or noisy

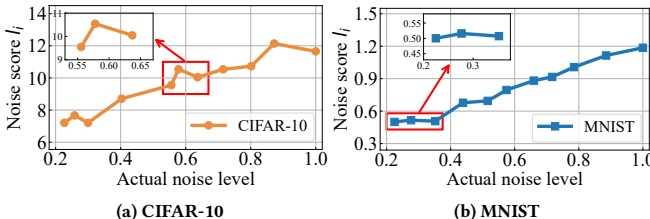

(a) CIFAR-10       (b) MNIST

**Figure 2: An illustration on the relationship between actual noise levels and the defined noise score on various datasets.**

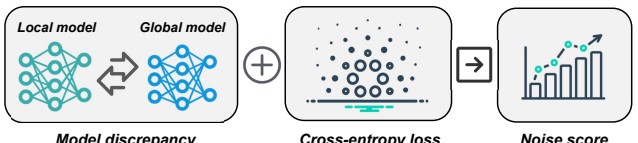

**Figure 3: The components of defined noise score.**

data [1, 15, 25]. Based on this evidence, they demonstrate the effectiveness of using Cross-Entropy (CE) loss to exhibit the data quality between noisy and clean labels. Following this insight, we let the noise level $l(\mathcal{D}_i)$ also be proportional to CE loss on each client's local model, i.e., $l(\mathcal{D}_i) \propto \text{CE}(y, \hat{y})$. As shown in Fig. 3, combining the analysis above yields the formal definition of the *noise score*,

$$l(\mathcal{D}_i) = -\|w_t - w_{i,t}\|^2 \sum_{(x_j, y_j) \in \mathcal{D}_i} y_j \log \psi_i(w_{i,t}, x_j), \forall i, t. \tag{12}$$

Here, $\psi_i(\cdot, \cdot)$ represents the learning model kept by client $i$, which can produce a predicted label $\hat{y}_j$ given a sample data $x_j$. This noise score has been evaluated using various datasets in a noisy FL setting, in which the noise is simulated using Guassian distributions. The results are depicted in Fig. 2, where x-axis represents the actual imposed noise level. It can be observed that the defined noise level $l(\mathcal{D}_i)$ precisely aligns with the actual noise level in most cases. This highlights the feasibility and accuracy of $l(\mathcal{D}_i)$. Note that, as marked by the red box, only a few cases exhibit inconsistency. However, this is acceptable, as we will further address and mitigate the noise issue in the subsequent phases of FPIN.

## 4.2 Noisy Client Selection

In order to enhance the performance of noisy FL settings, the key in the selection phase is to select as appropriate client sets with low noise and high contributions as possible. To this end, we have proposed in Definition 1 the statistical utility $u_{i,t}$ that measures both the data quality and noise level of clients. A simple method is to directly select the top-$K$ clients with the highest value of $u_{i,t}$. However, this method requires all clients to participate in the local training at each round and produces $u_{i,t}, \forall i \in [N], \forall t \in [T]$ cooperated with Server $\mathcal{S}$. This is impractical in real-world application scenarios because this method yields too much training cost and communication overhead, especially when total clients are sufficiently large.

As a result, we allow in this paper server $S$ to select the client set $I_t$ based on clients' previous utilities, like utility mean, instead of the last utility solicited from all clients at the current round. Clients just need to calculate their utilities when selected, thereby reducing a great deal of consumption. However, merely using the utility mean also raises a concern: several potential clients may not be selected all the time due to they does perform well at the former

**Algorithm 1: FPIN**

**Input:** Clients $[N]$, edge servers $[M]$, cardinality constraint $K$, time horizon $T$, cloud aggregation cycle $\tau$, learning ratio $\eta$, epoch number $E$, initial model $w_0$

**Output:** Global model $w_T$

1   $t \leftarrow 1$, $w_{i,1} \leftarrow w_0, \forall i \in [N]$. Select all clients once to generate noise score $l(\mathcal{D}_i)$ and statistical utilities $u_{i,1}, \forall i \in [N]$;

2   $U_{i,t} \leftarrow \{u_{i,t}\}, \forall i \in [N]$, $I_1 \leftarrow [N]$, $\mathcal{I} \leftarrow \{I_1\}$;

3   $t \leftarrow t + 1$, initialize $e_{i,t}$ based on Eq. 13;

4   **while** $t \leq T$ **do**

     // Cost submission phase

5     Clients submit costs $C_t = \{c_{i,t} : \forall i \in [N]\}$ to sever $\mathcal{S}$;

     // Client selection phase

6     Compute $\rho_{i,t} \leftarrow \bar{u}_{i,t} + e_{i,t}, \forall i \in [N]$ according to Eq. 13;

7     $I_t \leftarrow$ the top-$K$ clients with the highest value of $\rho_{i,t}/c_{i,t}$;

     // Local training phase

8     **foreach** client $i \in I_t$ **do**

9       **foreach** local epoch $\varsigma \in [1 : E]$ **do**

10        $w_{i,t-1} \leftarrow w_{i,t-1} - \eta_{t-1} \nabla F_i(w_{i,t-1}, d_i)$;

11      $w_{i,t} \leftarrow w_{i,t-1}$, $u_{i,t} \leftarrow \sqrt{|\mathcal{D}_i| \sum_{d \in \mathcal{D}_i} f_i(w_{i,t}, d)^2}/l(\mathcal{D}_i)$;

12      $U_{i,t} \leftarrow U_{i,t-1} \cup \{u_{i,t}\}$, update $e_{i,t+1}$;

13      Upload $w_{i,t}, u_{i,t}$ to server $\mathcal{S}$;

14     $u_{i,t} \leftarrow u_{i,t-1}$, $w_{i,t} \leftarrow w_{i,t-1}, \forall i \in [N] \setminus I_t$; $\mathcal{U}_t \leftarrow \{u_{i,t} : i \in [N]\}$;

     // Global aggregation phase

15     $w_t \leftarrow \sum_{i \in I_t} (|\mathcal{D}_i|/\sum_{k \in I_t} |\mathcal{D}_k|) w_{i,t}$; $w_{i,t} \leftarrow w_t, \forall i \in I_t$;

     // Payment determination phase

16     **foreach** client $i \in I_t$ **do** $p_{i,t} \leftarrow \pi_p(c_{i,t}, C_{-i,t}, \mathcal{U}_t, K)$;

17     $t \leftarrow t + 1$;

18   **return** The ultimate cloud model $w_T$

---

rounds. To address this concern, we modify the utility mean by including an additive term as follows:

$$\rho_{i,t} = \bar{u}_{i,t} + e_{i,t} \text{ and } e_{i,t} = \frac{c_{min} + u_{max}}{c_{min}} \sqrt{\frac{(K+1)\ln t}{|U_{i,t-1}|}}. \quad (13)$$

We then formally refer to $\rho_{i,t}$ as the *modified mean*. Here, $U_{i,t}$ is the statistical utility sequence presented in Definition 1 and $|U_{i,t-1}|$ represents the number of times client $i$ has been selected in the first $t-1$ rounds. Then, $\bar{u}_{i,t} = \sum_{u \in U_{i,t-1}} u/|U_{i,t-1}|$ is the empirical mean of client $i$'s statistical utilities, $c_{min} = \min_{i \in [N], t \in [T]} c_{i,t}$, $u_{max} = \min_{i \in [N], t \in [T]} u_{i,t}$, and $e_{i,t}$ is the exploration term. We can find that a client's modified mean will gradually increase over rounds if it is not selected consistently, i.e., $|U_{i,t-1}|$ remains unchanged, until this client is selected. This means term $e_{i,t}$ performs well in exploring potential clients.

We next provide a detailed description of our mechanism FPIN in Algorithm 1. We begin with initializing the model of each client with $w_0$ and selecting all clients once to update the necessary variables $u_{i,t}$, $U_{i,t}$, $e_{i,t}$, and $l(\mathcal{D}_i)$ for the first selection (lines 1-3). In the iterative part (lines 4-17), each communication round $t > 1$ primarily includes three phases. All clients reveal their costs $c_{i,t}$ to sever $\mathcal{S}$ at the cost submission phase. Then the top $K$ clients with the highest value of $\rho_{i,t}/c_{i,t}$ are selected at the client selection phase. At the local training phase, each selected client $i \in I_t$ trains its local

**Algorithm 2: Flexible pricing policy, i.e., $\pi_p$**

**Input:** Cardinality constraint $K$, selected client set $I_t$, cost set $C_{i,t}$, utility set $\mathcal{U}_t$, an arbitrary constant $\theta$

**Output:** Payment $p_{k,t}$ for client $k \in I_t$

1   Compute $\rho_{i,t}, \forall i \in [N]$ based on $C_{i,t}, \mathcal{U}_t$ according to Eq. 13;

2   Sort clients in descending order with respect to $\rho_{i,t}/c_{i,t}$;

3   Compute the critical value $p_{k,t}^c \leftarrow \frac{\rho_{k,t}}{\rho_{K+1,t}} c_{K+1,t}$;

4   Search a client $j$ satisfying that $\rho_{j,t}/c_{j,t} < \rho_{k,t}/c_{k,t}$ while $\rho_{j,t} > \rho_{k,t}$, and let $p'_{k,t} \leftarrow \frac{\rho_{k,t}}{\rho_{j,t}} c_{j,t}$, $\gamma \leftarrow 0$;

5   **while** $\rho_{j,t}/c_{j,t} < \rho_{k,t}/c_{k,t}$ **do**

6     **foreach** client $i \in [N]$ **do** $c_{i,t} \leftarrow c_{i,t} + \theta$;

7     Sort clients in descending order with $\rho_{i,t}/c_{i,t}$ again;

8     $\gamma \leftarrow \gamma + 1$;

9   $\gamma_0 \leftarrow \gamma$, $p'_{k,t} \leftarrow p'_{k,t} - (\gamma_0 - 1)(1 - \frac{\rho_{k,t}}{\rho_{j,t}})\theta$;

10   **return** $p_{k,t} = \min\{p'_{k,t}, p_{k,t}^c\}$

---

model, updates necessary variables, and uploads $w_{i,t}, u_{i,t}$ to server $\mathcal{S}$ (lines 8-13), while other clients $[N] \setminus I_t$ remain unchanged (line 14). At the aggregation phase, server $\mathcal{S}$ aggregates models from the selected clients using FedAVG [16] (line 15) and communicates the aggregated model $w_t$ back. Finally, server $S$ pays each selected client with payment $p_{i,t}$ decided using pricing policy $\pi_p$ (line 16), which will be described in the following section. These phases above run iteratively until round $T$, yielding final global model $w_T$.

## 4.3 Flexible Pricing

**Table 1: The frequency of finding client $j$**

| $N$ | $10^1$ | $10^2$ | $10^3$ | $10^4$ | $10^5$ |
|---|---|---|---|---|---|
| # find $j$ | 456 | 785 | 926 | 974 | 992 |
| # not find $j$ | 554 | 215 | 74 | 26 | 8 |
| Success rate | 45.6% | 78.5% | 92.6% | 97.4% | 99.2% |

In this section, we focus on designing a pricing policy $\pi_p$ in FPIN to prevent clients' strategic behaviors, compensate their expense, and incentivize them to clean noise spontaneously. The details of $\pi_p$ are described in Algorithm 2. Assuming to determine the payment for client $k$, we first sort all clients based on their mean-cost ratio $\rho_{i,t}/c_{i,t}$. Then a classic Myerson's critical value $p_{k,t}^c$ is derived for comparison, where $\rho_{K+1,t}$ and $c_{K+1}$ are the modified utility and cost of the $(K+1)$-th client in the sorted sequence, which is essentially the first client not selected by server $\mathcal{S}$. Next, we try to search a client $j$ with a lower mean-cost ratio $\rho_{j,t}/c_{j,t}$ and a higher modified mean $\rho_{j,t}$ compared to client $k$ (line 4). Table 1 illustrates that the success rate of finding such a client is satisfactory. We then initialize a temporary payment $p'_{k,t}$ as $(\rho_{k,t}/\rho_{j,t})c_{j,t}$ and a counter $\gamma$ as 0 for following calculation. When such a client $j$'s mean-cost ratio is less than that of client $k$, Algorithm 2 enters the iterative part (lines 5-8). Here, each client's cost is updated by adding the noise score $l(\mathcal{D}_i)$. We then re-sort clients in descending order and update counter $\gamma$ until client $j$'s mean-cost ratio becomes no less than that of client $k$. Finally, we get client $k$'s payment of $p_{k,t} = \min\{p'_{k,t}, p_{k,t}^c\}$.

# 5 THEORETICAL ANALYSIS

In this section, we start by proving the truthfulness, individual rationality, and noise robustness of the pricing policy $\pi_p$ in Theorems 1-3. Next, we provide a convergence analysis of FPIN in Theorem 4.

## 5.1 Analysis on Pricing

**THEOREM** 1. *The payment determined by $\pi_p$ for each client $i \in [N]$ achieves asymptotic truthfulness.*

PROOF. We begin by proving that the selection policy $\pi_s$ in Algorithm 1 (lines 6-7) is cost-monotonic. A selection policy is considered cost-monotonic [22] if, when a client $i$ is selected based on its mean-cost ratio $\rho_{i,t}/c_{i,t}$, this client will also be selected with a different mean-cost ratio $\rho_{i,t}/c'_{i,t}$, where $c'_{i,t} < c_{i,t}$. According to Algorithm 1, selecting client $i$ indicates that $\rho_{i,t}/c_{i,t} > \hat{\rho}_{K+1,t}/c_{K+1,t}$, where $\rho_{K+1,t}$ and $c_{K+1,t}$ are the modified mean and cost of the $(K + 1)$-th client in the sorted clients. If client $i$'s cost is decreased from $c_{i,t}$ to $c'_{i,t}$, it still holds that $\rho_{i,t}/c'_{i,t} > \hat{\rho}_{K+1,t}/c_{K+1}$. Thus, the selection policy $\pi_s$ in Algorithm 1 is cost-monotonic. Furthermore, we provide two cases to demonstrate the truthfulness of $\pi_s$.

**Case 1:** When $p_{k,t} = p^c_{k,t}$, we prove that $p^c_{k,t}$ ensures truthfulness. Assume that a client $i$ is selected when it truthfully submits its cost $c_{i,t}$, yielding a profit of $\beta = p^c_{i,t} - c_{i,t}$. If client $i$ submits a fake cost $c'_{i,t} \neq c_{i,t}$, there are two possible outcomes with the new mean-cost ratio $\rho_{i,t}/c'_{i,t}$. (1) Client $i$ is selected, and its profit is still $\beta'_1 = p^c_{i,t} - c_{i,t}$ as client $i$'s payment does not rely on its own cost $c_{i,t}$ according to Algorithm 2. (2) Client $i$ is not selected, resulting in a profit of $\beta'_2 = 0$. Note that we will prove $p^c_{i,t} - c_{i,t} > 0$ in Theorem 2. Therefore, we have $\beta > \max\{\beta'_1, \beta'_2\}$, indicating that client $i$ can maximize its profit by truthfully submitting its cost $c_{i,t}$.

Similarly, assume that a client $i$ is not selected when submitting its cost $c_{i,t}$ truthfully and its profit is $\beta = 0$ now. If client $i$ declares a fake cost $c'_{i,t}$, there are two possible outcomes. (1) Client $i$ is not selected, leading to a profit of $\beta'_1 = 0$. (2) Client $i$ is selected, and its profit is $\beta'_2 = p^c_{i,t} - c_{i,t}$ now. When client $i$'s mean-cost ratio changes from $\rho_{i,t}/c_{i,t}$ to $\rho_{i,t}/c'_{i,t}$, there must be a client whose mean-cost ratio is exceeded by client $i$. Without loss of generality, we assume that client is $j \in [N]$, and the following holds

$$\rho_{i,t}/c'_{i,t} \geq \rho_{K+1,t}/c_{K+1,t} \geq \rho_{j,t}/c_{j,t} > \rho_{i,t}/c_{i,t}. \quad (14)$$

Then, client $i$ gets the payment of

$$p^c_{i,t} = (\rho_{i,t}/\rho_{K+1,t})c_{K+1,t} < (\rho_{K+1,t}/\rho_{K+1,t})c_{i,t} = c_{i,t}. \quad (15)$$

We thus have $\beta > \max\{\beta'_1, \beta'_2\}$, meaning client $i$ achieves maximum profit by truthfully submitting its cost $c_{i,t}$.

**Case 2:** When $p_{k,t} = p'_{k,t}$, we prove that $p^c_{k,t}$ ensures asymptotic truthfulness. According to Definition 4 and Algorithm 2, a selected client $i \in I_t$ who submits the cost truthful will receive a payment of $p'_{i,t}$. If client $i$ is not truthful, the maximum payment it can receive is $p^c_{i,t}$ under the Myerson-based pricing strategy [19]. Consequently, we say that $p^c_{i,t} - p'_{i,t} = o(p'_{i,t})$ holds since, for any constant $\lambda$, we can find a constant $\epsilon = (p^c_{i,t} - c_{i,t})/\lambda$ such that $p^c_{i,t} - p'_{i,t} < \lambda p'_{i,t}, \forall p'_{i,t} > \epsilon$. This is because $p^c_{i,t} - p'_{i,t} < p^c_{i,t} - c_{i,t}$ due to Eq. 23 in Theorem 3. □

**THEOREM** 2. *The payment determined by $\pi_p$ for each client $i \in [N]$ achieves individual rationality.*

PROOF. Individual rationality indicates that each client $i \in [N]$ can obtain a payment $p_{i,t}$ that is no less than its cost $c_{i,t}$. In line 3 of Algorithm 2, we get the critical value $p^c_{k,t} = (\rho_{k,t}/\rho_{K+1,t})c_{K+1,t}$ for client $k$. Since client $k$ is selected among the top $K$ clients, we have $\rho_{k,t}/c_{k,t} \geq \rho_{K+1,t}/c_{K+1,t}$. It then follows that the critical value $p^c_{k,t} \geq c_{k,t}$ and we have $p'_{k,t} > c_{k,t}$ in Eq. 23. Thus, the payment ensures $p_{k,t} = \min\{p'_{k,t}, p^c_{k,t}\} \geq c_{k,t}$, which completes the proof. □

**THEOREM** 3. *The payment determined by $\pi_p$ for each client $i \in I_t$ achieves noise robustness, i.e., $p_{i,t} < c_{i,t} + \theta$.*

PROOF. As described in Algorithm 2 (line 4), we search a client $j$ satisfying that $\rho_{k,t}/c_{k,t} > \rho_{j,t}/c_{j,t}$. Then it holds that

$$p'_{k,t} = (\rho_{k,t}/\rho_{j,t})c_{j,t} > c_{k,t}. \quad (16)$$

We define a function $h(x) = \frac{\rho_{k,t}}{\rho_{j,t}}(c_{j,t} + x\theta) - (c_{k,t} + x\theta)$. Then,

$$h(0) = (\rho_{k,t}/\rho_{j,t})c_{j,t} - c_{k,t} > 0, \quad (17)$$

$$h(\gamma_0 - 1) = (\rho_{k,t}/\rho_{j,t})(c_{j,t} + (\gamma_0 - 1)\theta) - (c_{k,t} + (\gamma_0 - 1)\theta) > 0, \quad (18)$$

$$h(\gamma_0) = (\rho_{k,t}/\rho_{j,t})(c_{j,t} + \gamma_0\theta) - (c_{k,t} + \gamma_0\theta) \leq 0, \quad (19)$$

where $\gamma_0$ is the counter that indicates the termination of the while loop in line 5. The inequality in Eq. 17 holds due to Eq. 16, while Eqs. 18-19 hold since, according to Algorithm 2 (line 5), the while loop terminates when $\rho_{j,t}/(c_{j,t} + \gamma_0\theta) \geq \rho_{k,t}/(c_{k,t} + \gamma_0\theta)$. When $\gamma = \gamma_0 - 1$, the while loop does not terminate, and it holds that $\rho_{j,t}/(c_{j,t} + \gamma\theta) < \rho_{k,t}/(c_{k,t} + \gamma\theta)$. According to Eqs. 17-19, it follows

$$h(\gamma_0 - 1) = (\rho_{k,t}/\rho_{j,t})c_{j,t} - c_{k,t} - (1 - \rho_{k,t}/\rho_{j,t})(\gamma_0 - 1)\theta > 0, \quad (20)$$

$$h(\gamma_0) = (\rho_{k,t}/\rho_{j,t})c_{j,t} - c_{k,t} - (1 - \rho_{k,t}/\rho_{j,t})\gamma_0\theta \leq 0 \quad (21)$$

Therefore,

$$(1 - \frac{\rho_{k,t}}{\rho_{j,t}})(\gamma_0 - 1)\theta + c_{k,t} < \frac{\rho_{k,t}}{\rho_{j,t}}c_{j,t} \leq (1 - \frac{\rho_{k,t}}{\rho_{j,t}})\gamma_0\theta + c_{k,t}. \quad (22)$$

Due to the definition of $p'_{k,t}$ in Algorithm 3 (lines 3 and 9), we have

$$c_{k,t} < p'_{k,t} \leq (1 - \rho_{k,t}/\rho_{j,t})\theta + c_{k,t}. \quad (23)$$

Since $\rho_{j,t}, \rho_{k,t} > 0$ by Eq. 13, we have $p_{k,t} \leq p'_{k,t} < c_{k,t} + \theta$. By assigning a specific noise score $\theta$ such that $\theta \propto 1/l(\mathcal{D}_i)$, the payment determined by $\pi_p$ can ensure $p_{k,t} - c_{k,t} < \phi(1/l(\mathcal{D}_i))$, in which $\phi(\cdot)$ is a proportional function. This implies that a client with a lower noise level (i.e., a smaller noise score $\theta$) may receive more additional profit, thereby achieving noise robustness. □

## 5.2 Analysis on Convergence

We provide several assumptions and additional notations motivated by previous studies [12, 23] to analyze convergence of FPIN.

**ASSUMPTION** 1. *The objective function $F_i(\cdot), \forall i \in [N]$ is L-smooth, i.e., given any model pair, $w$ and $\varphi$, it holds that $F_i(\varphi) \leq F_i(w) + \langle \varphi - w, \nabla F_i(w) \rangle + L\|\varphi - w\|/2$.*

**ASSUMPTION** 2. *The objective function $F_i(\cdot), \forall i \in [N]$ is $\mu$-strongly convex, i.e., given any model pair, $w$ and $\varphi$, it holds that $F_i(\varphi) \geq F_i(w) + \langle \varphi - w, \nabla F_i(w) \rangle + \mu\|\varphi - w\|/2$.*

**ASSUMPTION** 3. *The objective function $F_i(\cdot), \forall i \in [N]$ is $\mathcal{L}$-Lipschitz continuous, i.e., given any model pair, $w$ and $\varphi$, it holds that $|F_i(\varphi) - F_i(w)| \leq \mathcal{L}\|\varphi - w\|$ and $\mathcal{L} > 0$.*

These assumptions regarding objective function $F_i(\cdot)$ are normal and regular, say logistic regression and softmax classifier. Moreover,

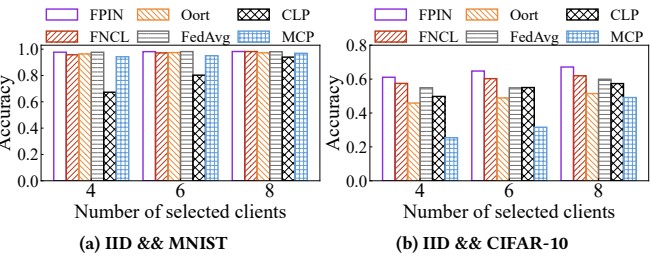

Figure 4: The accuracy of various selection policies based on different datasets given both Non-IID and IID scenarios.

**(a) IID & MNIST   (b) Non-IID & MNIST   (c) IID & CIFAR-10   (d) Non-IID & CIFAR-10**

Figure 5: The accuracy of various selection policies as the number of selected clients, $K$, is increased.

**(a) IID && MNIST   (b) IID && CIFAR-10**

we define several additional notations to accurately display the FL process. Since every communication round $t \in [T]$ comprises $E$ epochs (i.e., the local training phase in lines 7-14 of Algorithm 1 in main paper), we leverage $\varsigma \in [TE]$ to represent all involved epochs. When $\varsigma/E \in [T]$, it implies that $\varsigma$ is the end epoch within a round. The local training phase of each client is re-described as

$$\varphi_{i,\varsigma} \leftarrow w_{i,\varsigma-1} - \eta_{\varsigma-1}\nabla F_i(d_i, w_{i,\varsigma-1}), \tag{24}$$

$$w_{i,\varsigma} \leftarrow \begin{cases} \varphi_{i,\varsigma} & \text{if } \varsigma/E \notin [T], \\ \pi_a(\{\varphi_{i,\varsigma}, i \in I_\varsigma\}) & \text{if } \varsigma/E \in [T]. \end{cases} \tag{25}$$

Here, $I_\varsigma$ denotes the currently selected client set, i.e., $I_t$ where $t = \lceil \varsigma/E \rceil - 1$. $\pi_a$ is the aggregation policy using FedAvg. We denote the means of $\varphi_{i,\varsigma}$ and $w_{i,\varsigma}$ by $\bar{\varphi}_\varsigma = \sum_{i \in [N]} p_i \varphi_{i,\varsigma}$ and $\bar{w}_\varsigma = \sum_{i \in [N]} p_i w_{i,\varsigma}$ like settings in [12]. Let $g_\varsigma = \sum_{i \in [N]} p_i \nabla F_i(d_i, w_{i,\varsigma})$ and $\bar{g}_\varsigma = \sum_{i \in [N]} p_i \nabla F_i(w_{i,\varsigma})$, where $\nabla F_i(w_{i,\varsigma})$ is the expected gradient over full data of client $i$. Let $w^*$ represent the optimal parameter of the global model that maximizes $F(w^*)$ in Eq. 2. We then provide Theorem 4 regarding FPIN's convergence rate.

Please refer to Appendix for the convergence analysis of FPIN, i.e., Theorem 4.

## 6  SIMULATIONS

### 6.1  Simulation Settings

**Datasets and Models.** We perform all simulations for this paper using PyTorch on a workstation featuring an NVIDIA GeForce RTX 3090 GPU based on two widely recognized datasets, MNIST and CIFAR-10. We utilize two simple CNN models that incorporate batch normalization layers to implement FPIN. Each model consists of three blocks, with each block comprising a convolutional layer, a batch normalization layer, and a ReLU activation function. We then apply SGD optimizers, exponential decay learning rate schedulers, and cross-entropy loss functions in simulations. We also explore

non-IID scenario for FPIN, where client heterogeneity is accurately modeled using the Dirichlet distribution[11]. Dir(r) represents the proportions of each class allocated to each client, sampled from the Dirichlet distribution. By default, we distribute the entire dataset evenly among all clients.

**Benchmarks.** We evaluate the effectiveness of FPIN by comparing it with several well-known client selection policies.

(1) **FNCL**[25]: Federated Noisy Client Learning (FNCL) operates by identifying noisy clients through an accurate assessment of data quality and model divergence. To address the data heterogeneity introduced by these noisy clients, FNCL applies a robust layerwise aggregation method, which adaptively aggregates the local models from clients.

(2) **Oort**[7]: This is promising FL framework that employs a bandit-style strategy for client selection. Oort indirectly enhances the diversity of datasets in FL. We initialize Oort's exploration rate at 0.9, establish its minimum exploration rate at 0.1, and define its decay factor as 0.97.

(3) **FedAvg**[16]: FedAvg employs a random policy to selects clients straightforwardly at each round. It performs local training on these selected clients and aggregates clients' local models by weighting their data volume.

(4) **Loss and Cost Policy (CLP)** : A variant of the selection policy in FPIN. Considering that clients with low noise scores are beneficial for training, MCP re-designs the selection metric of clients based on two factors: clients' local loss and submitted costs, defined as $u_{i,t} = 1/(\text{loss}_{i,t} \cdot c_{i,t})$.

(5) **Minimum Cost Policy (MCP)** : Another variant of FPIN, where the utility only considers the bid as a factor, selecting the client with the minimum bid.

**Parameters.** Specifically, the number of participating clients is set to $N = 40$ and $K = 8$, with a total of $T = 150$ global rounds. SGD is implemented as the local training optimizer with a learning rate of 0.01, a local batch size of 64, and local training epoch $E = 5$ for all datasets. In Gaussian noise distribution with a mean of $\mu = 0.3$ and variance of $\sigma = 0.45$.

### 6.2  Simulation Results

**Effectiveness Evaluation.** We assess the performance of FPIN in comparison to other baselines in both IID and Non-IID settings, as illustrated in Fig. 4, under the conditions of noisy data from truncated Gaussian distributions. The experimental results demonstrate that FPIN outperforms the other baselines on the CIFAR-10 dataset Compared to other benchmarks, FPIN enhances accuracy by 5% to

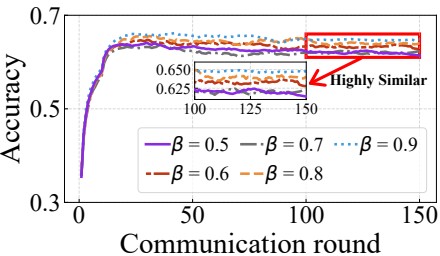

**Figure 6: Parameter evaluation of the noise detection in FPIN with Gaussian distribution.**

18% in the IID scenario and by 15% to 20% in the Non-IID scenario. During the training process, FPIN effectively mitigates the influence of noisy clients' models by accurately distinguishing between noisy and clean clients. The reliability score can effectively assess each client by evaluating the quality of their local model and training loss. Then, FPIN tends to select clients with low noise levels and high quality to enhance the model's performance. On the MNIST dataset, FPIN slightly outperforms other benchmark algorithms. We observe that the global model trained using MCP and MB fails to converge, as evidenced by the instability of their test accuracy curves towards the end of training in the Non-IID scenario. This instability arises because noisy clients steer the collaborative model's updates in a divergent direction during the model aggregation process.

**Selection Evaluation.** As shown in Fig. 5a and 5b, we observe that as the number of selected clients increases, the accuracy of the global model also improves. However, due to differences in the difficulty of the datasets, the accuracy gap after convergence on MNIST is not significant. Oort combines top-k statistical utility sampling with random exploration to select clients. However, it cannot promptly adjust the selected client set, as clients chosen in the previous round have a higher probability of being selected again in the next round. Furthermore, the inherent randomness in client selection for FedAvg, MCP, and even FNCL contributes to the suboptimal performance of the global model. In the presence of noise, while the selection of clients of CLP with low training loss mitigates some of the noise's impact, it also discards potentially valuable clients, namely those with high training loss. Due to the concentrated client selection of CLP, the limited amount of data fitted does not optimize performance.

After applying this process, we classify the clients into high-noise clients and low-noise clients. This method allows us to accurately identify low-noise clients in the subsequent selection phase of federated learning, thereby improving overall efficiency and effectiveness. As shown in Fig. 6, notably the parameter $\beta$ governs the sensitivity of the noisy clients, and reducing its value may lead to a decrease in the accuracy of detecting these noisy clients. Interestingly, we observed that $\beta$ exhibited minimal sensitivity in the presence of Gaussian noise.

**Individual Rationality of FPIN.** FP can flexibly determine payments for each winner based on a factor that accounts for the level of client noise. As shown in Fig. 7a, clients with lower noise levels receive higher payment ceilings, while clients with higher noise levels have payments closer to the $y = x$ line. Meanwhile, this also indicates that the profit obtained by each client is non-negative. Then, the Cumulative Distribution Function (CDF) of the profits

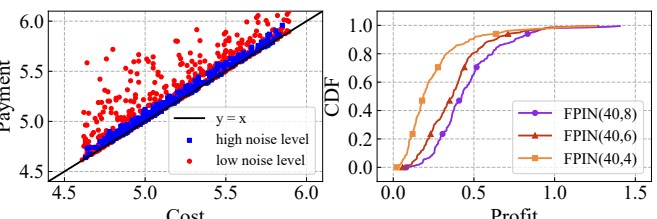

**Figure 7: Costs and payments of clients with different noise levels (Left part). CDF of clients' profits based on increasing numbers of selected clients (right part).**

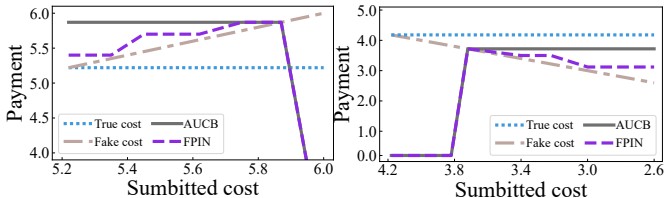

**Figure 8: Payments of a winner client (left part) and a loser client (right part) as their submitted cost are varied.**

for winners is shown in Fig. 7b. Our analysis reveals that all profits generated using FPIN are non-negative, indicating that FPIN meets the criteria for individual rationality, as demonstrated in Theorem 4. As the number of selected clients increases, the total profit per round naturally rises.

**Truthfulness of FPIN.** Additionally, we validate the performance in terms of truthfulness. In Fig. 8a, we observe that as the declared bid increases, the winner continues to be selected, and the FPIN payment increases until it reaches the maximum value, which corresponds to the AUCB payment. However, when the bid exceeds the critical value of 5.87, the winner is no longer selected, meaning no payment is made. Thus a client will not increase its bid since it may make the client not pulled. In Fig. 8b, we discover that the loser is initially not selected, resulting in a payment of zero. As the bid decreases below 3.72, the loser will be selected. However, the client incurs a negative profit, meaning its payment does not cover the actual cost. Therefore, clients has no incentive to misreport their costs.

## 7 CONCLUSION

In this paper, we closely investigate a method for detecting the level of noisy data from clients in federated learning and propose a selection strategy that prioritizes clients with low noise and high contribution. Additionally, we develop an accurate and flexible pricing mechanism that incentivizes clients to clean their noisy data while enforcing truthfulness. These approaches form FPIN, with its effectiveness validated through both theoretical analysis and numerical simulations. Simulation results demonstrate that FPIN significantly improves the performance of global model with noisy clients in both homogeneous and heterogeneous federated learning settings, while also ensuring individual rationality and asymptotic truthfulness.

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

# A APPENDICES

**THEOREM 4.** *Given Assumptions 1-3, the following holds*

$$\mathbb{E}[F(w_T)] - F(w^*) \le \frac{\mathcal{L}}{TE+\kappa}\left(\frac{\lambda_1+\lambda_2}{4\mu^2}+(\kappa+1)C_1\right), \quad (26)$$

*where* $\kappa = \max\{E, 8L/\mu - 1\}$, $\lambda_1 = 4L\Gamma(M) + 16(E-1)^2\mathcal{G}^2$, $\lambda_2 = (NK^2E^2\mathcal{G}^2 + 4N^2\Delta_s^2 \ln(1.25/\delta))/K^4$, $\mathcal{G}$ *is a constant defined by [23], and* $C_1 = \mathbb{E}[\|\bar{w}_1 - w^*\|^2]$. *The learning ratio is set to* $\eta_\varsigma = 2/(\mu(\varsigma+\kappa))$, *where* $\varsigma$ *represents the* $\varsigma$*-th epoch.*

**PROOF.** The convergence property essentially reflects the discrepancy in objective functions between the realized model $w$ and the optimal model $w^*$. It is hard to directly obtain this discrepancy, so we analyze the difference between models,

$$\|\bar{w}_\varsigma - w^*\|^2 = \|\bar{w}_\varsigma - \bar{\varphi}_\varsigma + \bar{\varphi}_\varsigma - w^*\|^2$$
$$\le 2(\|\bar{w}_\varsigma - \bar{\varphi}_\varsigma\|^2 + \|\bar{\varphi}_\varsigma - w^*\|^2), \quad (27)$$

where the last inequality holds due to the Cauchy-Schwarz inequality. Afterward, we separately bound $\|\bar{w}_\varsigma - \bar{\varphi}_\varsigma\|^2$ and $\|\bar{\varphi}_\varsigma - w^*\|^2$ in steps 1-2, and bound Eq. 27 in step 3.

**Step 1**: Bounding $\|\bar{w}_\varsigma - \bar{\varphi}_\varsigma\|^2$. When $\varsigma/E \notin [T]$, it holds that $\bar{w}_\varsigma = \bar{\varphi}_\varsigma$ due to their definitions. When $\varsigma/E \in [T]$,

$$\mathbb{E}[\|\bar{w}_\varsigma - \bar{\varphi}_\varsigma\|^2] = \mathbb{E}[\|(1/K)\sum_{i\in I_\varsigma}\varphi_{i,\varsigma} - \bar{\varphi}_\varsigma\|^2]$$
$$= (1/K^2)\mathbb{E}[\|\sum_{i\in[N]}\mathbb{I}\{i\in I_\varsigma\}(\varphi_{i,\varsigma} - \bar{\varphi}_\varsigma)\|^2]$$
$$\le (N/K^2)\mathbb{E}_{I_\varsigma}[\sum_{i\in[N]}\mathbb{I}\{i\in I_\varsigma\}\|\varphi_{i,\varsigma} - \bar{\varphi}_\varsigma\|^2]$$
$$= (N/K^2)\sum_{i\in[N]}\Pr\{i\in I_\varsigma\}\|\varphi_{i,\varsigma} - \bar{\varphi}_\varsigma\|^2. \quad (28)$$

The first two equalities follow from definitions of $\bar{w}_\varsigma$ and $\bar{\varphi}^2$. The first inequality holds also due to the Cauchy-Schwarz inequality, i.e., $\|\sum_{i\in[N]}(x_i - y_i)\| \le \sum_{i\in[N]}\|x_i - y_i\|$. Let $\varsigma_0 = \varsigma - E$. Then epoch $\varsigma_0$ is the communication round recalling that $\varsigma/E \in [T]$. This implies all clients have the identical model $w_{i,\varsigma_0}, \forall i \in [N]$. Then,

$$\text{Eq. 28} \le (N/K^2)\sum_{i\in[N]}\|(\varphi_{i,\varsigma} - \bar{w}_{\varsigma_0}) - (\bar{\varphi}_\varsigma - \bar{w}_{\varsigma_0})\|^2$$
$$\le (N/K^2)\sum_{i\in[N]}\|(\varphi_{i,\varsigma} - \bar{w}_{\varsigma_0})\|^2$$
$$= (N/K^2)\sum_{i\in[N]}2(\|(\varphi_{i,\varsigma} - \bar{w}_{\varsigma-1}) + \cdots + (\bar{w}_{\varsigma_0+1} - \bar{w}_{\varsigma_0})\|^2$$
$$\le (NE/K^2)\sum_{i\in[N]}\sum_{\tau\in[\varsigma_0+1,\varsigma]}2\|\eta_{\tau-1}\nabla F_i(d_i, w_{i,\tau-1})\|^2, \quad (29)$$

The second inequality holds from $\mathbb{E}[\|x - \mathbb{E}[x]\|^2] \le \mathbb{E}[\|x\|^2]$ and the third inequality follows from the Cauchy-Schwarz inequality similarly. Further, due to Theorem 2.2 in [23], the expected squared norm of stochastic gradients is upper bounded by a constant $\mathcal{G}$, i.e., $\mathbb{E}[\|\nabla F_i(d_i, w_{i,\varsigma})\|^2] \le \mathcal{G}^2$. Therefore, it holds that

$$\mathbb{E}[\|\bar{w}_\varsigma - \bar{\varphi}_\varsigma\|^2] \le \eta_{\varsigma_0}^2 NE^2\mathcal{G}^2/K^2 + 2(N^2/K^2)(2\Delta_s^2/(\epsilon^2K^2))\cdot$$

$$\ln(1.25/\delta) \le (\eta_{\varsigma-1}^2 NK^2E^2\mathcal{G}^2 + 4N^2\Delta_s^2 \ln(1.25/\delta)/\epsilon^2)/(2K^4). \quad (30)$$

The last inequality holds since the learning rate $\eta_\varsigma$ is set to be non-increasing and $\eta_{\varsigma_0} \le 2\eta_{\varsigma-1}$ as in [24].

**Step 2**: Bounding $\|\bar{\varphi}_\varsigma - w^*\|^2$. Without out loss of generality, we set $\varsigma \leftarrow \varsigma + 1$ and bound $\|\bar{\varphi}_{\varsigma+1} - w^*\|^2$ for convenience of writing. Then, the following holds,

$$\|\bar{\varphi}_{\varsigma+1} - w^*\|^2 = \|\bar{w}_\varsigma - \eta_\varsigma g_\varsigma - w^* - \eta_e \bar{g}_\varsigma + \eta_\varsigma \bar{g}_\varsigma\|^2 \tag{31}$$

$$\leq 2(\|\bar{w}_\varsigma - w^* - \eta_\varsigma \bar{g}_\varsigma\|^2 + \|\eta_\varsigma \bar{g}_\varsigma - \eta_\varsigma g_\varsigma\|^2)$$

$$= 2(\|\bar{w}_\varsigma - w^*\|^2 - 2\eta_\varsigma \langle \bar{w}_\varsigma - w^*, \bar{g}_\varsigma \rangle + \eta_\varsigma^2 \|\bar{g}_\varsigma\|^2 + \eta_\varsigma^2 \|\bar{g}_\varsigma - g_\varsigma\|^2),$$

where the first inequality holds similarly with Eq. 27. For term $2\eta_\varsigma \langle \bar{w}_\varsigma - w^*, \bar{g}_\varsigma \rangle$ in Eq. 31, it holds that

$$\langle \bar{w}_\varsigma - w^*, \bar{g}_\varsigma \rangle = \sum_{i \in [N]} p_i \langle \bar{w}_\varsigma - w^*, \nabla F_i(w_{i,\varsigma}) \rangle$$

$$= \sum_{i \in [N]} p_i (\langle \bar{w}_\varsigma - w_{i,\varsigma}, \nabla F_i(w_{i,\varsigma}) \rangle + \langle w_{i,\varsigma} - w^*, \nabla F_i(w_{i,\varsigma}) \rangle)$$

$$\geq \sum_{i \in [N]} p_i ((1/4\eta_\varsigma) \|\bar{w}_\varsigma - w_{i,\varsigma}\|^2 + \eta_\varsigma \|\nabla F_i(w_{i,\varsigma})\|^2) +$$

$$\sum_{i \in [N]} p_i (F_i(w_{i,\varsigma}) - F_i(w_i^*) + (\mu/8) \|w_{i,\varsigma} - w^*\|^2), \tag{32}$$

where the last inequality is by using AM-GM inequality for $\langle \bar{w}_\varsigma - w_{i,\varsigma}, \nabla F_i(w_{i,\varsigma}) \rangle$ and $\mu$-strongly convexity of $F_i(\cdot)$ for $\langle w_{i,\varsigma} - w^*, \nabla F_i(w_{i,\varsigma}) \rangle$. For term $\|\bar{g}_\varsigma\|^2$, it holds that $\|\bar{g}_\varsigma\|^2 \leq \sum_{i \in [N]} p_i \|\nabla F_i(w_{i,\varsigma})\|^2 \leq 2L \sum_{i \in [N]} p_i (F_i(w_{i,\varsigma}) - F_i^*)$, where the first inequality follows from the Cauchy-Schwarz inequality and the second is by applying $L$-smoothness of $F_i(\cdot)$ in Assumption 1. As for term $\|\bar{g}_\varsigma - g_\varsigma\|^2$,

$$\|\bar{g}_\varsigma - g_\varsigma\|^2 = \|\sum_{i \in [N]} p_i (\nabla F_i(d_i, w_{i,\varsigma}) - \nabla F_i(w_{i,\varsigma}))\|^2 \tag{33}$$

$$\leq \sum_{i \in [N]} N p_i^2 \|\nabla F_i(d_i, w_{i,\varsigma}) - \nabla F_i(w_{i,\varsigma})\|^2 \leq \sum_{i \in [N]} N p_i^2 \varrho_i^2.$$

The first inequality follows from the Cauchy-Schwarz inequality and the second is by the variance bound on stochastic gradients for client $i$ [23], $\mathbb{E}[\|\nabla F_i(d_i, w_{i,\varsigma}) - \nabla F_i(w_{i,\varsigma})\|^2] \leq \varrho_i^2$. Combining these inequalities yields

$$\text{Eq. } 31 \leq 2(((1 - \eta_\varsigma \mu)/4) \|\bar{w}_\varsigma - w^*\|^2 + \sum_{i \in [N]} N \eta_e^2 p_i^2 \varrho_i^2 +$$

$$2\eta_\varsigma (2L\eta_\varsigma - 1) \sum_{i \in [N]} p_i (F_i(w_{i,\varsigma}) - F_i(w_i^*))). \tag{34}$$

The inequality follows from $\sum_{i \in [N]} \|w_{i,\varsigma} - w^*\|^2 \geq N \|\bar{w}_\varsigma - w^*\|^2$ in our settings and $L$-smoothness of $F_i(\cdot)$. We proceed to bound the term $\mathcal{F} = \sum_{i \in [N]} p_i (F_i(w_{i,\varsigma}) - F_i(w_i^*))$ in Eq. 34. Considering that $2L\eta_\varsigma - 1 < 0$, we then have

$$\mathcal{F} = \sum_{i \in [N]} p_i (F_i(w_{i,\varsigma}) - F_i(\bar{w}_\varsigma)) + \sum_{i \in [N]} p_i (F_i(\bar{w}_\varsigma) - F_i(w_i^*))$$

$$\geq -\sum_{i \in [N]} p_i (L\eta_\varsigma (F_i(\bar{w}_\varsigma) - F_i(w_i^*)) + 1/(2\eta_\varsigma) \|w_{i,\varsigma} - \bar{w}_\varsigma\|^2)$$

$$+ F(\bar{w}_\varsigma) - F(w^*) = (1 - \eta_\varsigma L) \sum_{i \in [N]} p_i (F_i(\bar{w}_\varsigma) - F(w^*)) -$$

$$L\eta_\varsigma \sum_{i \in [N]} p_i (F(w^*) - F_i(w_i^*)) - 1/(2\eta_\varsigma) \sum_{i \in [N]} \|w_{i,\varsigma} - \bar{w}_\varsigma\|^2$$

$$\geq L\eta_\varsigma \Gamma - (1/(2\eta_\varsigma))(4\eta_\varsigma^2 (E-1)^2 \mathcal{G}^2). \tag{35}$$

The first inequality follows from the convexity of $F_i(\cdot)$, the $L$-smoothness, and the AM-GM inequality. We define the individual discrepancy of clients as $\Gamma = F^* - \sum_{i \in [N]} p_i F_i^*$ that reflects the non-IIDness of their local datasets. In addition, the second inequality also follows from the fact of $\sum_{i \in [N]} \|w_{i,\varsigma} - \bar{w}_\varsigma\|^2 \leq 4\eta_\varsigma^2 (E-1)^2 \mathcal{G}^2$ obtained similarly to Eq. 30 and the fact of $1 - \eta_\varsigma L > 0$, $\sum_{i \in [N]} p_i (F_i(\bar{w}_\varsigma) - F(w^*)) = F(\bar{w}_\varsigma) - F(w^*) > 0$. Further,

$$\|\bar{\varphi}_{\varsigma+1} - w^*\|^2 \leq ((1 - \eta_\varsigma \mu)/2) \|\bar{w}_\varsigma - w^*\|^2 + \eta_\varsigma^2 \mathcal{B}, \tag{36}$$

where $\mathcal{B} = 4L\Gamma + 16(E-1)^2 \mathcal{G}^2$. The inequality is by Eq. 34-35 and $\eta_\varsigma \in (0, 1/(4L))$ and $2\eta_\varsigma (1 - 2L\eta_\varsigma) \in [\eta_\varsigma, 2\eta_\varsigma)$.

**Step 3**: Combining results of Steps 1-2 (Eqs. 30, 36) yields

$$\mathbb{E}[\|\bar{w}_{\varsigma+1} - w^*\|^2] \leq (1 - \eta_\varsigma \mu) \mathbb{E}[\|\bar{w}_\varsigma - w^*\|^2] + 2\eta_\varsigma^2 \mathcal{B} + \tag{37}$$

$$2\eta_\varsigma^2 N K^2 E^2 \mathcal{G}^2 / (2K^4) = (1 - \eta_\varsigma \mu) \cdot \mathbb{E}[\|\bar{w}_\varsigma - w^*\|^2] + \eta_\varsigma^2 (\lambda_1 + \lambda_2).$$

Here, $\lambda_2 = NE^2 \mathcal{G}^2 / K^2$ and $\lambda_1 = 2\mathcal{B}$. So given a time-varying learning rate $\eta_\varsigma = 2/\mu(\varsigma + \kappa)$, where $\kappa = \max\{E, 8L/\mu - 1\}$ such that $\eta_1 \leq 1/(4L)$ and $\eta_\varsigma \leq 2\eta_{\varsigma+E}$. Setting $C_\varsigma = \mathbb{E}[\|\bar{w}_{\varsigma+1} - w^*\|^2]$, we prove by induction that

$$C_\varsigma \leq \xi_i / (\varsigma + \kappa), \; \xi_i = \max\{(\lambda_1 + \lambda_2)/(4\mu^2), (\kappa+1)C_1\}. \tag{38}$$

When $\varsigma = 1$, Eq. 38 holds due to the definition of $\xi_i$. Assuming that Eq. 38 holds at epoch $\varsigma$, it also holds for $\varsigma + 1$,

$$C_{\varsigma+1} \leq (1 - \eta_\varsigma \mu) C_\varsigma + \eta_\varsigma^2 (\lambda_1 + \lambda_2)$$

$$\leq (1 - \frac{2}{\varsigma + \kappa}) \frac{\xi_i}{\varsigma + \kappa} + \frac{4(\lambda_1 + \lambda_2)}{\mu^2 (\varsigma + \kappa)^2} = \frac{\xi_i (\varsigma + \kappa - 2)}{(\varsigma + \kappa)^2} + \frac{4(\lambda_1 + \lambda_2)}{\mu^2 (\varsigma + \kappa)^2}$$

$$= \frac{\xi_i (\varsigma + \kappa - 1)}{(\varsigma + \kappa)^2} + \frac{4(\lambda_1 + \lambda_2) - \mu^2 \xi_i}{\mu^2 (\varsigma + \kappa)^2} \leq \frac{\xi_i}{\varsigma + \kappa + 1}. \tag{39}$$

The last inequality is by the fact of $\xi_i \geq (\lambda_1 + \lambda_2)/(4\mu^2)$ and $(\varsigma + \kappa - 1)/(\varsigma + \kappa)^2 \leq (\varsigma + \kappa - 1)/((\varsigma + \kappa)^2 - 1)$. Finally,

$$\mathbb{E}[F(w_T)] - F(w^*) \leq \frac{\mathcal{L}\xi_i}{TE + \kappa} \leq \frac{\mathcal{L}}{TE + \kappa} (\frac{\lambda_1 + \lambda_2}{4\mu^2} + (\kappa+1)C_1).$$

The first inequality holds by the $\mathcal{L}$-Lipschitz continuity of both $F_i(\cdot)$ and $F(\cdot)$, and the second one is by the definition of $\xi_i$. Further, we can represent $\mathbb{E}[F(w_T)] - F(w^*)$ asymptotically as $O(N\mathcal{G}^2/(TK^2))$, where $\mathcal{G}$ is closely related to the total noise level of clients. $\square$

Theorem 4 implies that even affected by noisy data issue, FPIN still achieves a sublinear convergence rate that scales as $O(1/T)$. This means the discrepancy $\mathbb{E}[F(w_T)] - F(w^*)$ approaches 0 as $T$ becomes sufficiently large. Moreover, and a smaller total number of clients $N$, a lower total noise level $\mathcal{G}$, and an increased number of selected clients $K$ will yield to a converged training performance, which is reasonable and consistent with practical reality.

