# OpenReview forum: "Dealing with Noisy Data in Federated Learning: An Incentive Mechanism with Flexible Pricing"
_ACM.org/TheWebConf/2025/Conference — WWW 2025 Poster_

### Official Review · Reviewer_g12J · 2024-11-27

**Novelty:** 5
**Technical Quality:** 5

**Review:**

This paper introduces FPIN, a framework addressing noisy data in federated learning via noise detection, client selection, and dynamic pricing.
Pros:1. The experimental results and theoretical analysis provide strong support for its effectiveness. 2.Proposes an innovative pricing mechanism to incentivize client participation and noise reduction.
Cons:1.The datasets are relatively small and not enough 2. The computational costs are not clearly analyzed.

**Questions:**

1.Have you considered testing the framework on diverse datasets (e.g., text data) to demonstrate its generalizability across different domains?
2.What is the computational overhead of the proposed method in real-world scenarios, particularly for devices with limited resources?

**Reviewer Confidence:**

3: The reviewer is confident but not certain that the evaluation is correct

**Scope:**

3: The work is somewhat relevant to the Web and to the track, and is of narrow interest to a sub-community

---

### Official Review · Reviewer_CHmM · 2024-12-01

**Novelty:** 5
**Technical Quality:** 5

**Review:**

The manuscript presents a detailed analysis of the convergence behavior in a federated learning framework under the presence of noisy gradients, with a focus on the sublinear convergence rate. The theoretical foundation is built upon sound assumptions such as smoothness, strong convexity, and bounded gradient variance, and the authors successfully derive rigorous bounds on the expected squared norm of the model parameters. The work's originality lies in its careful treatment of gradient noise, client heterogeneity, and time-varying learning rates, contributing to the body of literature on federated learning convergence.

**Strengths:**
- The analysis is thorough, providing a clear and mathematically sound argument for sublinear convergence in noisy federated learning setups.
- The manuscript makes a meaningful contribution by addressing the effect of non-IID data on convergence, a crucial issue in federated learning.
- The derivations are well-organized, and the proof structure is clear, especially in the use of inductive reasoning for bounding the error.
- The use of a time-varying learning rate adds realism and practical applicability to the results, as real-world federated systems often adapt learning rates over time.

**Weaknesses:**
- The presentation could benefit from further clarification in some places, especially in the transition between steps of the proof. Some terms are introduced without sufficient explanation, which may hinder understanding for readers less familiar with the topic.
- While the manuscript provides an asymptotic analysis of convergence, the practical implications and how the theoretical results translate to real-world scenarios are not sufficiently discussed. Including some experimental validation or insights into how the derived bounds hold in practice would strengthen the work.
- The notation in some parts could be streamlined to improve readability. For example, the various constants introduced in the proofs (such as \(\lambda_1\), \(\lambda_2\), and \(\xi_i\)) could be explained more clearly in terms of their relevance to the final bounds.
- The manuscript assumes that the gradient variance \(\mathcal{G}\) is bounded, which is reasonable in theory, but it would be useful to discuss how sensitive the convergence rate is to variations in \(\mathcal{G}\) in practice. More discussion of this would be valuable, particularly for real-world datasets with varying noise levels.

**Questions:**

1. The manuscript assumes that the gradient variance \(\mathcal{G}\) is bounded. How sensitive is the convergence rate to changes in the magnitude of \(\mathcal{G}\)? Could the results hold in the case of highly noisy gradients, or is further noise regularization required for convergence?

2. Given the theoretical results, could you provide more practical insights or simulations to demonstrate how the convergence bounds behave in real-world federated learning systems? Specifically, how do the results hold for heterogeneous data and varying client participation?

3. Some constants like \(\lambda_1\), \(\lambda_2\), and \(\xi_i\) appear frequently in the proof but are not always explained in terms of their role in the final convergence bound. Could you clarify how these constants impact the practical significance of the final results? How do they relate to the overall error reduction?

4. The learning rate is set to decrease over time. Could you discuss the effect of alternative learning rate schedules (e.g., constant or adaptive learning rates) on the convergence behavior? Would the same sublinear convergence rate hold under different learning rate policies?

5. In the analysis, the number of selected clients \(K\) plays a key role in the convergence bounds. Could you elaborate on how the convergence rate is affected by client selection strategies in federated learning? Are there any practical considerations or limitations when selecting clients based on data quality or computational resources?

6. The analysis accounts for non-IID data using the discrepancy term \(\Gamma\). How does the model perform when the data heterogeneity is extreme (e.g., when clients have significantly different distributions)? Are there specific settings where the convergence analysis might fail or require modification?

7. The inductive proof for convergence (Eq. 39) suggests that the error decreases at a sublinear rate. Can you provide more intuition or an explanation for why this is guaranteed under noisy gradients and varying client updates? How does the performance scale with the number of clients \(N\) or the number of selected clients \(K\)?

8. The work presents a theoretical bound for convergence, but practical federated learning often faces issues like stragglers, network latency, and hardware heterogeneity. How would these factors affect the results, and are there any considerations for handling such challenges in federated learning systems?

9. The manuscript assumes that clients’ updates are synchronized. What happens if there is staleness in the client updates, as is common in federated settings? Would the convergence analysis still hold in the presence of stale gradients?

**Reviewer Confidence:**

3: The reviewer is confident but not certain that the evaluation is correct

**Scope:**

4: The work is relevant to the Web and to the track, and is of broad interest to the community

---

### Official Review · Reviewer_u9vt · 2024-12-01

**Novelty:** 4
**Technical Quality:** 4

**Review:**

This paper presents a novel method for solving the data noise issue in the reverse-auction-based system. Specifically, the author proposed a noise detection policy, client selection policy, and pricing policy. The noise level is determined by the model discrepancy and cross-entropy loss. The top k clients with the highest mean-cost ratio are selected for global model aggregation. The pricing policy produces payment according to the client's noise level.

pros:
- The paper focuses on important data noise problems in FL and proposes an interesting incentive mechanism to tackle the problem.
- The paper is well-structured and easy to follow. The figures are clear and well-documented.

cons:
-  The method proposed in this paper is based on the statement. "Pre-simulations revel that, during the training process, clients with high noise levels consistently exhibit a local model that diverges more significantly from the global model compared to low-noise and clean clients." However, without considering the heterogeneity of both the hardware and training settings among clients, e.g., different GPU, different batch sizes, learning rates, and epochs ..., the model discrepancy cannot represent the real noise level in the dataset.
- The proposed method claims to have the advantage of detecting the noise level without the auxiliary dataset. However, as I mentioned in my previous point, model discrepancy and cross-entropy loss could fail to represent the noise of the data in the heterogeneous setting. So, it feels like the method prevents the client from changing the model dramatically rather than detecting the actual noise in the dataset. In such a case, you can always benefit from involving the auxiliary dataset to detect the noise level.
- The model and dataset used for evaluation are simple. It would be better to consider different deep learning tasks, model architectures, and federated learning settings instead of just a basic CNN and the MNIST dataset.
- The paper claims to address the issue of data noise and encourages clients to clean their data. However, it does not provide any guidance on how clients can effectively clean the data. To me, the incentive appears to simply encourage clients to "change the model parameters as little as possible."

**Questions:**

- In Figure 2, how is the actual noise level calculated?
- Do different model architectures have varying effects on detecting noise levels?
- This method avoids the use of auxiliary datasets, claiming that it "incurs additional training costs and reduces generality." Can you show how much training cost is saved by the proposed method?
- How does the proposed method ensure consistent noise level detection across different model architectures using the same dataset?
- If the model is randomly initialized, then the clients with good data quality will still produce a large model discrepancy, right? In this case, how can the server decide who has better data quality?

**Reviewer Confidence:**

3: The reviewer is confident but not certain that the evaluation is correct

**Scope:**

3: The work is somewhat relevant to the Web and to the track, and is of narrow interest to a sub-community